# SCONE: Surface Coverage Optimization in Unknown Environments by Volumetric Integration

**Antoine Guédon**     **Pascal Monasse**     **Vincent Lepetit**
LIGM, Ecole des Ponts, Univ Gustave Eiffel, CNRS, France
`{antoine.guedon,pascal.monasse,vincent.lepetit}@enpc.fr`

## Abstract

Next Best View computation (NBV) is a long-standing problem in robotics, and consists in identifying the next most informative sensor position(s) for reconstructing a 3D object or scene efficiently and accurately. Like most current methods, we consider NBV prediction from a depth sensor like Lidar systems. Learning-based methods relying on a volumetric representation of the scene are suitable for path planning, but have lower accuracy than methods using a surface-based representation. However, the latter do not scale well with the size of the scene and constrain the camera to a small number of poses. To obtain the advantages of both representations, we show that we can maximize surface metrics by Monte Carlo integration over a volumetric representation. In particular, we propose an approach, SCONE, that relies on two neural modules: The first module predicts occupancy probability in the entire volume of the scene. Given any new camera pose, the second module samples points in the scene based on their occupancy probability and leverages a self-attention mechanism to predict the visibility of the samples. Finally, we integrate the visibility to evaluate the gain in surface coverage for the new camera pose. NBV is selected as the pose that maximizes the gain in total surface coverage. Our method scales to large scenes and handles free camera motion: It takes as input an arbitrarily large point cloud gathered by a depth sensor as well as camera poses to predict NBV. We demonstrate our approach on a novel dataset made of large and complex 3D scenes.

## 1   Introduction

Next Best View computation (NBV) is a long-standing problem in robotics [6, 29], which consists in identifying the next most informative sensor position(s) for reconstructing a 3D object or scene efficiently and accurately. Typically, a position is evaluated on how much it can increase the total coverage of the scene surface. Few methods have relied on Deep Learning (DL) for the NBV problem, even though DL can provide useful geometric prior to obtain a better prediction of the surface coverage [33, 16, 25]. Like most current methods, we consider NBV prediction from a depth sensor. Existing methods based on a depth sensor rely either on a volumetric or on a surface-based representation of the scene geometry. Volumetric mapping-based methods can compute collision efficiently, which is practical for path planning in real case scenarios [20, 23, 24, 14, 1, 7]. However, they typically rely on voxels or a global embedding [12, 3, 21, 22, 27, 5] for the scene, which results in poor accuracy in reconstruction and poor performance in NBV selection for complex 3D objects. On the contrary, surface mapping-based methods that process directly a dense point cloud of the surface as gathered by the depth sensor are efficient for NBV prediction with high-detailed geometry. They are however limited to very specific cases, generally a single, small-scale, isolated object with the camera constrained to stay on a sphere centered on the object [15, 24, 8, 4, 13, 14, 14, 33, 16, 25]. Thus, they cannot be applied to the exploration of 3D scenes.

36th Conference on Neural Information Processing Systems (NeurIPS 2022).

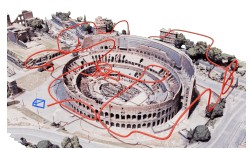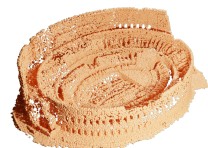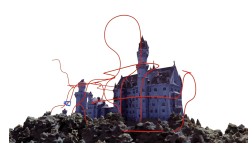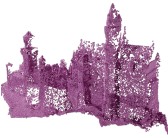

Figure 1: **Our Next Best View (NBV) method SCONE can handle unknown large-scale 3D scenes** to produce accurate 3D reconstruction inside a given 3D bounding box. Here, we call SCONE iteratively within a naive path planning algorithm to compute a complete camera trajectory avoiding collisions and obtain a complete 3D model. Despite being trained only on ShapeNet 3D models, it generalizes to complex scenes as shown above. (input 3D model courtesy of Brian Trepanier, under CC License. Downloaded from Sketchfab.)

As shown in Figure 1, we introduce a volumetric DL method that efficiently identifies NBVs for unknown large-scale 3D scenes in which the camera can move freely. Instead of representing the scene with a single global embedding, we choose to use a grid of local point clouds, which scales much better to large and complex 3D scenes. We show how to learn to predict the visibility of unseen 3D points in all directions given an estimate of the 3D scene geometry. We can then integrate these visibilities in the direction of any camera by using a Monte Carlo integration approach, which allows us to optimize the camera pose to find the next most informative views. We call our method SCONE, for Surface Coverage Optimization in uNknown Environments.

In this respect, we introduce a theoretical framework to translate the optimization of surface coverage gain, a surface metric on manifolds that represents the ability of a camera to increase the visible area of the surface, into an optimization problem on volumetric integrals. Such a formalism allows us to use a volumetric mapping of geometry, which is convenient not only to scale the model to exploration tasks and scene reconstruction, but also to make probabilistic predictions about geometry.

In particular, given a partial point cloud gathered by a depth sensor, our model learns to make predictions with virtually infinite resolution about the occupancy probability in the scene volume by learning a deep implicit function [17, 28, 18, 30, 31, 19]. Such predictions scale to very large point clouds since they depend only on neighborhood geometry. Then, our model leverages a self-attention mechanism [26] to predict occlusions and compute informative functions mapped on a continuous sphere that represent visibility likelihood of points in all directions. The occupancy probability field is finally used as a basis to sample points and compute Monte Carlo integrals of visibility scores.

Since NBV learning-based methods are mostly limited to single, small-scale, centered object reconstruction in literature, we first compare the performance of our model to the state of the art on the ShapeNet dataset [2], following the protocol introduced in [33]. While our method was designed to handle more general frameworks such as 3D scene reconstruction and continuous cameras poses in the scene, it outperforms the state of the art for dense reconstruction of objects when the camera is constrained to stay on a sphere centered on the object. We then conduct experiments in large 3D environments using a simple planning algorithm that builds a camera trajectory online by iteratively selecting NBVs with SCONE. Since, to the best of our knowledge, we propose the first supervised Deep Learning method for such free 6D motion of the camera, we created a dataset made of several large-scale scenes under the CC License for quantitative evaluation. We made our code and this dataset available for allowing comparison of future methods with SCONE on our project webpage: `https://github.com/Anttwo/SCONE`.

## 2 Approach

Let us consider a depth sensor exploring a 3D scene, at time step $t \geq 0$. Using its observations at discrete time steps $j$ with $0 \leq j \leq t$, the sensor has gathered a cloud of points distributed on the surface of the scene. We refer to this cloud as the *partial surface point cloud*, as it describes the part of the surface seen –or *covered*– by the sensor in the scene. To solve the NBV problem, we want to identify a camera pose that maximizes the coverage of previously unseen surface.

To this end, our method takes as input the partial surface point cloud as well as the history of 6D camera poses at time steps $j \leq t$ (*i.e.* all previous positions and orientations of the sensor). Our

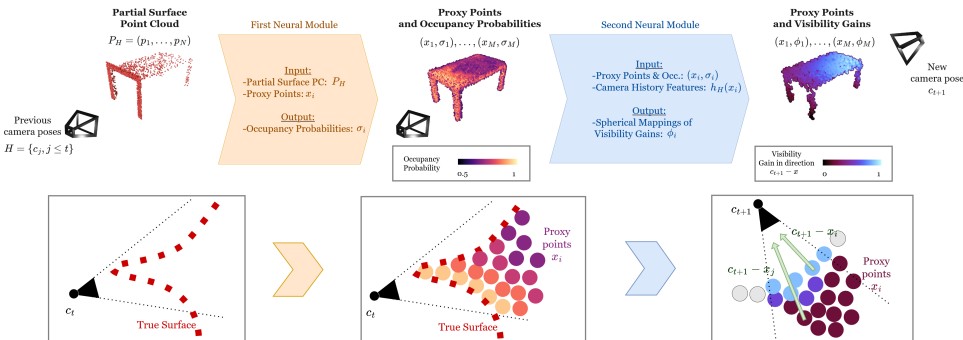

Figure 2: **The main steps of our method SCONE**. At time $t$, the depth sensor has visited camera poses $H = \{c_j, j \leq t\}$ and gathered a partial surface point cloud $P_H$ on the true surface, shown in red in the left image. Using its first neural module, our model predicts from the real surface points $P_H$ an occupancy probability distribution over proxy points $(x_1, ..., x_N)$ shown in the middle. The points $(x_1, ..., x_N)$ are sampled uniformly in the scene; we refer to them as *proxy points* because we use them to encode the volume. For readability, the figure does not show the proxy points with an occupancy value under $0.5$. To compute the coverage gain of any next camera pose $c_{t+1}$, the model samples a subset of proxy points $x_i$ in the field of view of $c_{t+1}$ and uses its second module to predict the visibility gain in direction $c_{t+1} - x_i$ for each point $x_i$, as shown on the right. The proxy points are sampled with probabilities proportional to their occupancy value. Moreover, for each proxy point $x_i$, the second module encodes the relative positions of previous cameras with specific features $h_H(x_i)$. We finally integrate visibility gains over proxy points in the field of view of $c_{t+1}$ to approximate the volumetric coverage gain integral appearing in Equation 6.

approach is built around two successive steps, each relying on a dedicated neural module as shown in figure 2: First, we make a prediction about the geometry of the scene, to estimate where the uncovered points could be. Then, we predict the visibility gain of uncovered points from any new camera pose; The NBV is finally selected as the camera with the most new visible points in its field of view.

Although we seek to maximize a surface metric such as surface coverage gain, our method relies on a volumetric representation of the object or scene to reconstruct. In this regard, we show that we can maximize a surface metric by integrating over a volumetric representation with virtually infinite resolution. As we argue below, such a representation is not only useful for collision-free robot navigation but is also much more efficient for optimizing surface coverage gain than the alternative of identifying the 3D points lying on the surface, which is difficult in an unknown and occluded environment. More exactly, we derive a volumetric integral that is asymptotically proportional to the surface coverage gain metric, which is enough for our maximization problem.

In the following subsection, we first present this derivation considering volume represented as a perfect binary occupancy map. We then present the two neural modules of SCONE and explain how we use them to predict all terms involved in the volumetric integral, by leveraging neural models and self-attention mechanisms to predict occupancy and occlusions.

## 2.1 Maximizing Surface Coverage Gain on a Binary Occupancy Map

Here, we consider a binary occupancy map $\sigma : \mathbb{R}^3 \to \{0, 1\}$ representing the volume of the target object or scene. We will relax our derivations to a probabilistic occupancy map when looking for the next best view in the next subsections. From the binary map $\sigma$, we can define the set $\chi$ of occupied points, *i.e.*, the set of points $x$ verifying $\sigma(x) = 1$, its surface as the boundary $\partial\chi$, and the surface coverage $C(c)$ achieved by a camera pose $c = (c_{\text{pos}}, c_{\text{rot}}) \in \mathcal{C} := \mathbb{R}^3 \times SO(3)$ as the following surface integral:

$$C(c) = \frac{1}{|\partial\chi|_S} \int_{\partial\chi} v_c(x)\, \mathrm{d}x, \tag{1}$$

where $|\partial\chi|_S := \int_{\partial\chi} \mathrm{d}x$ is the area of surface $\partial\chi$. $\chi_c \subset \chi$ is the subset of occupied points contained in the field of view of camera $c$, and $v_c(x)$ is the visibility of point $x$ from camera $c$, *i.e.*, $v_c(x) = \mathbb{1}_{\chi_c}(x) \cdot \mathbb{1}\left(\sigma\left(\{(1-\lambda)c_{\text{pos}} + \lambda x \text{ such that } \lambda \in [0, 1)\}\right) = \{0\}\right)$.

Since we want to maximize the total coverage of the surface by all cameras during reconstruction, we are actually interested in maximizing the coverage of previously unobserved points rather than the absolute coverage. Given a set of previous camera poses, which we call the *camera history* $H \subset \mathcal{C}$, and a 3D point $x$, we introduce the *knowledge indicator* $\gamma_H : \mathbb{R}^3 \to \{0, 1\}$ such that $\gamma_H(x) = \max\{v_c(x) : c \in H\}$. We then define the *coverage gain* $G_H(c)$ of camera pose $c$ as:

$$G_H(c) = \frac{1}{|\partial\chi|_S} \int_{\partial\chi} \nu_c^H(x) \, \mathrm{d}x, \tag{2}$$

where $\nu_c^H(x) = (1 - \gamma_H(x)) \cdot v_c(x)$ is the *visibility gain* of $x$ in $\chi_c$, for camera history $H$. This function is equal to 1 iff $x$ is visible at pose $c$ but was not observed by any camera pose in $H$. Given a camera history $H$, our goal is to identify a pose $c$ that maximizes $G_H(c)$.

Given an occupancy map $\sigma$, we could evaluate the integral in Eq. (2) by simply sampling points $p$ on surface $\partial\chi$. However, in practice we will estimate the occupancy map iteratively in an unknown environment, and we will only have access to an occupancy probability distribution. Extracting surface points from such a probabilistic occupancy map gives results that can differ a lot from the true surface: Indeed, in 3D, a surface acts as a very concentrated set with zero-measure, and requires high confidence to give meaningful results. Instead of extracting surface points, we extend the properties of such points to a small spherical neighborhood of the surface. This will allow us to replace the maximization of a surface metric by the maximization of a volumetric integral, which is much easier to compute from our volumetric representation.

More exactly, we assume there exists a quantity $\mu_0 > 0$ such that any volume point in the spherical neighborhood $T(\partial\chi, \mu_0) := \{p \in \mathbb{R}^3 \mid \exists x \in \partial\chi, \|x - p\|_2 < \mu_0\}$ keeps the same visibility property as its neighboring surface points. With such a hypothesis, we give a thickness to the surface, which makes sense when working with discrete points sampled in space to approximate a volume.

To this end, we introduce a new visibility gain function $g_c^H$ to adapt the definition of the former visibility gain $\nu_c^H$. For any $0 < \mu < \mu_0$:

$$g_c^H(\mu; x) = \begin{cases} 1 & \text{if } \exists x_0 \in \partial\chi, \lambda < \mu \text{ such that } x = x_0 + \lambda N(x_0) \text{ and } \nu_c^H(x_0) = 1, \\ 0 & \text{otherwise}, \end{cases} \tag{3}$$

where $N$ is the inward normal vector field. With further regularity assumptions about the surface that are detailed in the appendix, such quantities are well defined. Assuming $\mu_0$ is small enough, the following explicit formula translates the surface approach into a volume integral for any camera pose $c \in \mathcal{C}$ and $\mu < \mu_0$:

$$\int_{T(\partial\chi,\mu)} g_c^H(\mu; x)\mathrm{d}x = \int_{\partial\chi} \int_{-\mu}^{\mu} g_c^H(\mu; x_0 + \lambda N(x_0)) \det(I - \lambda W_{x_0}) \, \mathrm{d}\lambda \, \mathrm{d}x_0, \tag{4}$$

with $W_{x_0}$ the Weingarten map at $x_0$, that is, the Hessian of the signed distance function on the boundary of $\chi$, which is continuous on the scene surface, assumed to be compact [10].

By developing the determinant, we find that $\det(I - \lambda W_{x_0}) = 1 + \lambda b(\lambda, x_0)$ where $b$ is a bounded function on the compact space $[-\mu, \mu] \times \partial\chi$. Moreover, for all $x_0 \in \partial\chi$, we have by definition $g_c^H(\mu; x_0 + \lambda N(x_0)) = g_c^H(\mu; x_0) = \nu_c^H(x_0)$ when $0 \leq \lambda < \mu$, and $g_c^H(\mu; x_0 + \lambda N(x_0)) = 0$ when $-\mu < \lambda < 0$. It follows that, for every $0 < \mu < \mu_0$:

$$\begin{aligned} \int_{T(\partial\chi,\mu)} g_c^H(\mu; x)\mathrm{d}x &= \int_{\partial\chi} \int_0^{\mu} g_c^H(\mu; x_0)(1 + \lambda b(\lambda, x_0)) \, \mathrm{d}\lambda \, \mathrm{d}x_0 \\ &= \mu \int_{\partial\chi} g_c^H(\mu; x_0) \, \mathrm{d}x_0 + \int_{\partial\chi} \int_0^{\mu} \lambda g_c^H(\mu; x_0) b(\lambda, x_0) \, \mathrm{d}\lambda \, \mathrm{d}x_0 \\ &= \mu |\partial\chi|_S G_H(c) + \int_{\partial\chi} \int_0^{\mu} \lambda g_c^H(\mu; x_0) b(\lambda, x_0) \, \mathrm{d}\lambda \, \mathrm{d}x_0. \end{aligned} \tag{5}$$

The complete derivations are given in the appendix.

Function $g_c^H(\mu; \cdot)$ is naturally equal to 0 for every point outside $T(\partial\chi, \mu)$. Moreover, considering the regularity assumptions we made on the compact surface, if $\mu_0$ is chosen small enough then for all $x_0 \in \partial\chi, \mu < \mu_0$, the point $x_0 + \mu N(x_0)$ is located inside the volume, such that $\int_{T(\partial\chi,\mu)} g_c^H(\mu; x) \, \mathrm{d}x = \int_\chi g_c^H(\mu; x) \, \mathrm{d}x$. Since $|g_c^H(\mu; \cdot)| \leq 1$ for all $c \in \mathcal{C}$ and $\mu > 0$, we deduce the following theorem by bounding $|b|$ on $[-\mu, \mu] \times \partial\chi$:

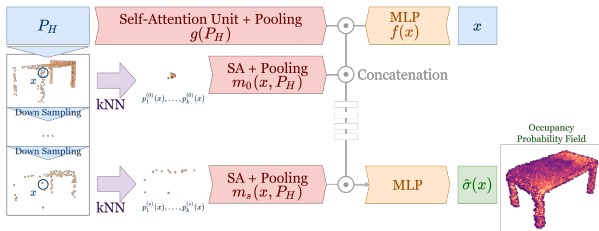

Figure 3: **Architecture of the first module of SCONE**, which predicts the occupancy probability field $\hat{\sigma}$. This module predicts the occupancy probability of a point $x$ from several inputs: The point $x$ after transformation by an MLP $f(x)$; A coarse global encoding $g(P_H)$ of the point cloud obtained by applying self-attention units on the sequence of points followed by a pooling operation; Multi-scale neighborhood features $m_i(x, P_H), i = 0, ..., N$ computed by down-sampling multiple times the point cloud, and encoding the $k$ nearest neighbors of $x$ with self-attention units after each sampling.

**Theorem 1.** *Under the previous regularity assumptions on the volume $\chi$ of the scene and its surface $\partial \chi$, there exist $\mu_0 > 0$ and $M > 0$ such that for all $\mu < \mu_0$, and any camera $c \in \mathcal{C}$:*

$$\left| \frac{1}{|\chi|_V} \int_\chi g_c^H(\mu; x) dx - \mu \frac{|\partial \chi|_S}{|\chi|_V} G_H(c) \right| \leq M\mu^2 , \tag{6}$$

*where $|\chi|_V$ is the volume of $\chi$.*

This theorem states that, asymptotically for small values of $\mu$, the volume integral $\int_\chi g_c^H(\mu; x) \, \mathrm{d}x$ gets proportional to the surface coverage gain values $G_H(c)$ that we want to maximize. This result is convenient since a volume integral can be easily approximated with Monte-Carlo integration on the volume and a uniform dense sampling based on the occupancy function $\sigma$. Consequently, the more points we sample in the volume, the smaller $\mu$ we can choose, and the closer maximizing the volume integral of spherical neighborhood visibility gain gets to maximizing the surface coverage gain.

### 2.2 Architecture

To approximate the volumetric integral in Equation 6 for any camera pose $c$, we need to compute $\chi_c$ as well as function $g_c^H$. In this regard, we need to compute both the occupancy map and the visibility gains of points for any camera pose. Since the environment is not perfectly known, we predict each one of these functions with a dedicated neural module. The first module takes as input the partial point cloud gathered by the depth sensor to predict the occupancy probability distribution. The second module takes as input a camera pose, a feature representing camera history as well as a predicted sequence of occupied points located in the camera field of view to predict visibility gains.

**Predicting the occupancy probability field $\hat{\sigma}$.** The occupancy function $\sigma$ is not known perfectly in practice. To represent occupancy, most volumetric NBV methods rely on memory-heavy representations (like an occupancy 3D-grid or a volumetric voxelization), that are generally less efficient for encoding fine details and optimizing dense reconstructions, and will necessarily downgrade the resolution compared to a point cloud directly sampled on the surface. To address this issue while still working with a volumetric representation of the scene, we use a deep implicit function to encode the 3D mapping of occupancy efficiently. Such a function has a virtually infinite resolution, and prevents us from saving a large 3D grid in memory. We thus approximate $\sigma$ with the first module of our model, which consists of a neural network $\hat{\sigma} : \mathcal{P}(\mathbb{R}^3) \times \mathbb{R}^3 \to [0, 1]$ that takes as inputs a partial surface point cloud $P_H \subset \mathbb{R}^3$ and query points $x \in \mathbb{R}^3$, and outputs the occupancy probability $\hat{\sigma}(P_H; x)$ of $x$. $P_H$ is obtained by merging together all depth maps previously captured from cameras in $H$.

As shown in Figure 3, rather than using a direct encoding of the global shape of $P_H$ as input to $\hat{\sigma}$, we take inspiration from [9] to achieve scalability and encode $P_H$ using features computed from the points' neighborhoods. The difference with [9] is that we rely on a multiscale approach: For a given query 3D point $x$, these features are computed from the $k$ nearest neighbors of $x$ computed at different scales. For each scale $s$, we downsample point cloud $P_H$ around $x$ into a sparser point cloud $P_H^{(s)}$ before recomputing the nearest neighbors $p_i^{(s)}(x), i = 1, ..., k$ of $x$: In this way, the size of the neighborhood increases with scale.

Next, for each value of $s$, we use small attention units [26, 11] on the sequence of centered neighborhood points $(p_1^{(s)}(x) - x, ..., p_k^{(s)}(x) - x)$ and apply pooling operations to encode each sequence of $k$ neighbors into a single feature that describes the local geometry for the corresponding scale. We finally concatenate these different scale features with another uncentered global feature as well as the query point $x$, and feed them to an MLP to predict the occupancy probability. The last global feature aims to provide really coarse information about the geometry and the location of $x$ in the scene.

This model scales well to large scenes: Adding points from distant views to the current partial point cloud does not change the local state of the point cloud. To avoid computing neighborhoods on the entire point cloud when reconstructing large scenes, we partition the space into cells in which we store the points in $P_H$. Given a query point $x$, we only use the neighboring cells to compute $p_i^{(s)}(x)$.

**Predicting the visibility gain $g_c^H$.** To maximize surface coverage gain, we need to compute the volumetric integral of visibility gain functions $g_c^H$. We do this again by Monte Carlo sampling, however, in unknown environments we cannot compute explicitly occlusions to derive visibility gain functions $g_c^H$ since the geometry, represented as a point cloud, is partially unknown and sparser than a true surface. We thus train the second module of our model to predict visibility gain functions by leveraging a self-attention mechanism that helps to estimate occlusion effects in the point cloud $P_H$.

In particular, for any camera pose $c \in \mathcal{C}$ and 3D point $x \in \chi_c$, the second module derives its prediction of visibility gains from three core features: (i) The predicted probability $\hat{\sigma}(P; x)$ of $x$ to be occupied, (ii) the occlusions on $x$ by the subvolume $\chi_c$ and (iii) the camera history $H$. To feed all this information to our model in an efficient way, we follow the pipeline presented in Figure 4. The model starts by using the predicted occupancy probability function $\hat{\sigma}$ to sample 3D points in the volume $\chi$. These samples will be used for Monte Carlo integration. We refer to these points as *proxy points* as we use them to encode the volume in the camera field of view, *i.e.*, in a pyramidal frustum view. We write $\hat{\chi}$ as the discrete set of sampled proxy points, and $\hat{\chi}_c$ as the set of proxy points located in the field of view of the camera $c$.

We first encode these proxy points individually by applying a small MLP on their 3D coordinates and their occupancy probability value concatenated together. Then, our model processes the sequence of these encodings with a self-attention unit to handle occlusion effects of subvolume $\chi_c$ on every individual point. Note there is no pooling operation on the output of this unit: The network predicts per-point features and does not aggregate predictions, since we do it ourselves with Monte Carlo integration. Next, for each proxy point $x \in \hat{\chi}$, we compute an additional feature $h_H(x)$ that encodes the history of camera positions $H$ with respect to this point as a spherical mapping: It consists in the projection on a sphere centered on $x$ of all camera positions for which $x$ was in the field of view. These features are concatenated to the outputs of the self-attention unit.

Our model finally uses an MLP on these features to predict the entire visibility gain functions of every point $x$ as a vector of coordinates in the orthonormal basis of spherical harmonics. With such a formalism, the model is able to compute visibility gains for points inside a subvolume in all directions with a single forward pass. In this regard, we avoid unnecessary computation and are able to process a large number of cameras in the same time when they share the same proxy points in their field of view (*e.g.*, reconstruction of a single object centered in the scene, where $\hat{\chi}_c = \hat{\chi}$ for all $c$, or when several cameras observe the same part of the 3D scene, *i.e.*, $\hat{\chi}_c = \hat{\chi}' \subset \hat{\chi}$ for several $c$).

Formally, if we denote by $Y_l^m : S^2 \rightarrow \mathbb{R}$ the real spherical harmonic of rank $(l, m)$ and $\phi_l^m(\hat{\chi}_c; x, h_H(x))$ the predicted coordinate of rank $(l, m)$ for proxy point $x \in \hat{\chi}_c$ with attention to subset $\hat{\chi}_c$ and camera history feature $h_H(x)$, the visibility gain of point $x$ in direction $d \in S^2$ is defined as

$$\sum_{l,m} \phi_l^m(\hat{\chi}_c; x, h_H(x)) \cdot Y_l^m(d) \tag{7}$$

so that the coverage gain $G_H(c)$ of a camera pose $c \in \mathcal{C}$ is proportional to

$$I_H(c) := \frac{1}{|\hat{\chi}|} \sum_{x \in \hat{\chi}} \left[ \mathbb{1}_{\hat{\chi}_c}(x) \sum_{l,m} \phi_l^m(\hat{\chi}_c; x, h_H(x)) \cdot Y_l^m \left( \frac{x - c_{pos}}{\|x - c_{pos}\|_2} \right) \right]. \tag{8}$$

The next best view among several camera positions is finally defined as the camera pose $c^*$ with the highest value for $I_H(c^*)$. Equation 8 is a Monte Carlo approximation of the volumetric integral in

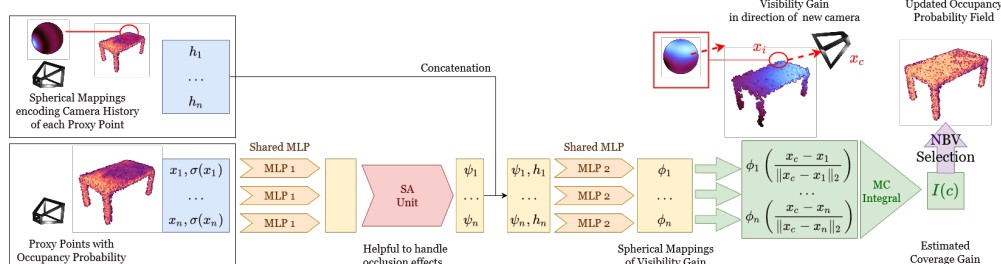

Figure 4: **Architecture of the second module of SCONE**, which predicts a visibility gain for each proxy point. To make this prediction, the model encodes the proxy points $x$ concatenated with their occupancy probability $\hat{\sigma}(x)$. We use an attention mechanism to take into account occlusion effects in the volume between the proxy points and their consequences on the visibility gains.

Equation 6, where the occupancy map and the visibility gains are predicted with neural networks. We choose to use a Monte-Carlo integral rather than a neural aggregator because this approach is simple, fast, makes training more stable, has good performance, better interpretability, and can handle sequences of arbitrary size. In particular, it implicitly encourages our model to compute meaningful visibility gains for each point since there is no asymmetry between the points.

We also use spherical harmonics to encode the camera history encoding $h_H(x)$ of each point $x$, which makes it a homogeneous input to the predicted output. Consequently, this input comes at the end of the architecture, and aims to adapt the visibility gain according to previous camera positions. This convenient representation, inspired by [32], allows us to handle free camera motion; On the contrary, several models in the literature encode the camera directions on a discrete sphere [33, 16, 25].

**Training.** We train the occupancy probability module alone with a Mean Squared Error loss with the ground truth occupancy map. We do not compute ground-truth visibility gains to train the second module since it would make computation more difficult and require further assumptions: We supervise directly on $I_H(c)$ by comparing it to the ground-truth surface coverage gain for multiple cameras, with softmax normalization and Kullback-Leibler divergence loss. Extensive details about the training of our modules and the choices we made are given in the appendix.

## 3 Experiments

As discussed in the introduction, deep learning-based NBV models for dense reconstruction are currently limited to single, small-scale, centered object reconstruction. To compare SCONE to these previous methods in this context, we first constrain the camera pose to lie on a sphere centered on an object. We then introduce our dataset made of 13 large-scale 3D models that we created to evaluate SCONE on free camera motions (3D models courtesy of Brian Trepanier, Andrea Spognetta, and 3D Interiors, under CC License; all models were downloaded from the website Sketchfab).

### 3.1 Next Best View for Single Object Reconstruction

We first compare the performance of our model to the state of the art on a subset of the ShapeNet dataset [2] introduced in [33] and following the protocol of [33]: We sample 4,000 training meshes from 8 specific categories of objects, 400 validation meshes and 400 test meshes from the same categories, and 400 additional test meshes from 8 categories unseen during training.

The evaluation on the test datasets consists of 10-view reconstructions of single objects. Given a mesh in the dataset, camera positions are discretized on a sphere. We start the reconstruction process by selecting a random camera pose, then we iterate NBV selection 9 times in order to maximize coverage with a sequence of 10 views in total. The evaluation metric is the area under the curve (AUC) of surface coverage throughout the reconstruction. This criterion not only evaluates the quality of final surface coverage, but also the convergence speed toward a satisfying coverage. Results are presented in Table 1. Further details about the evaluation, the estimation of ground truth surface coverage gains, and the metric computation are available in the appendix.

Table 1: **AUCs of surface coverage for several NBV selection methods for dense object reconstruction**, as computed on the ShapeNet test dataset following the protocol of [33], and after averaging over multiple seeds in the case of our method. For this experiment, we constrain the camera to stay on a sphere centered on the objects in order to compare with previous methods. Even if our model is designed to scale to entire scene reconstructions with free camera motion, it is still able to beat other methods trained for the specific case of dense object reconstruction with constrained camera motion.

| | Categories seen during training | | | | | | | | |
|---|---|---|---|---|---|---|---|---|---|
| Method | Airplane | Cabinet | Car | Chair | Lamp | Sofa | Table | Vessel | Mean |
| Random | 0.745 | 0.545 | 0.542 | 0.724 | 0.770 | 0.589 | 0.710 | 0.674 | 0.662 |
| Proximity Count [8] | 0.800 | 0.596 | 0.591 | 0.772 | 0.803 | 0.629 | 0.753 | 0.706 | 0.706 |
| Area Factor [23] | 0.797 | 0.585 | 0.587 | 0.751 | 0.801 | 0.627 | 0.725 | 0.714 | 0.698 |
| NBV-Net [16] | 0.778 | 0.576 | 0.596 | 0.743 | 0.791 | 0.599 | 0.693 | 0.667 | 0.680 |
| PC-NBV [33] | 0.799 | 0.612 | **0.612** | **0.782** | 0.800 | 0.640 | 0.760 | 0.719 | 0.716 |
| SCONE (Ours) | **0.827** | **0.625** | 0.591 | **0.782** | **0.819** | **0.662** | **0.792** | **0.734** | **0.729** |

| | Categories not seen during training | | | | | | | | |
|---|---|---|---|---|---|---|---|---|---|
| Method | Bus | Bed | Bookshelf | Bench | Guitar | Motorbike | Skateboard | Pistol | Mean |
| Random | 0.609 | 0.619 | 0.695 | 0.795 | 0.795 | 0.672 | 0.768 | 0.614 | 0.694 |
| Proximity Count | 0.646 | 0.645 | **0.749** | 0.829 | 0.854 | 0.705 | 0.828 | 0.660 | 0.740 |
| Area Factor | 0.629 | 0.631 | 0.742 | 0.827 | 0.852 | 0.718 | 0.799 | 0.660 | 0.732 |
| NBV-Net | 0.654 | 0.628 | 0.729 | 0.824 | 0.834 | 0.710 | 0.825 | 0.645 | 0.731 |
| PC-NBV | 0.667 | 0.662 | 0.740 | **0.845** | 0.849 | **0.728** | 0.840 | 0.672 | 0.750 |
| SCONE (Ours) | **0.694** | **0.689** | 0.746 | 0.832 | **0.860** | **0.728** | **0.845** | **0.717** | **0.764** |

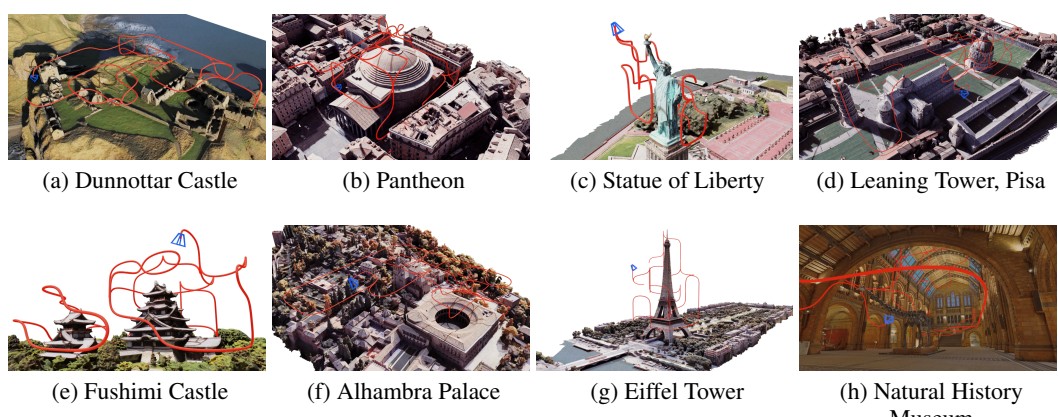

(a) Dunnottar Castle    (b) Pantheon    (c) Statue of Liberty    (d) Leaning Tower, Pisa

(e) Fushimi Castle    (f) Alhambra Palace    (g) Eiffel Tower    (h) Natural History Museum

Figure 5: **Application of our NBV approach to the reconstruction of large 3D structures**. The path, constructed iteratively in real time with our method, is shown in red (3D models courtesy of Brian Trepanier, Andrea Spognetta, and 3D Interiors, under CC License; all models were downloaded from the website Sketchfab).

## 3.2 Active View Planning in a 3D Scene

To evaluate the scalability of our model to large environments as well as free camera motion in 3D space, we also conducted experiments using a naive planning algorithm that incrementally builds a path in the scene: We first discretize the camera poses in the scene on a 5D grid, corresponding to coordinates $c_{pos} = (x_c, y_c, z_c)$ of the camera as well as the elevation and azimuth to encode rotation $c_{rot}$. The number of poses depends on the dimensions of the bounding box of the scene. This box is an input to the algorithm, as a way for the user to tell which part of the scene should be reconstructed. In our experiments, the number of different poses is around 10,000. Note we did not retrain our model for such scenes and use the previous model trained on ShapeNet as explained in Section 3.1.

The depth sensor starts from a random pose (from which, at least, a part of the main structure to reconstruct is visible). Then, at each iteration, our method estimates the coverage gain of direct neighboring poses in the 5D grid, and selects the one with the highest score. The sensor moves to this position, captures a new partial point cloud and concatenates it to the previous one. Since we focus on online path planning for dense reconstruction optimization and designed our model to be scalable

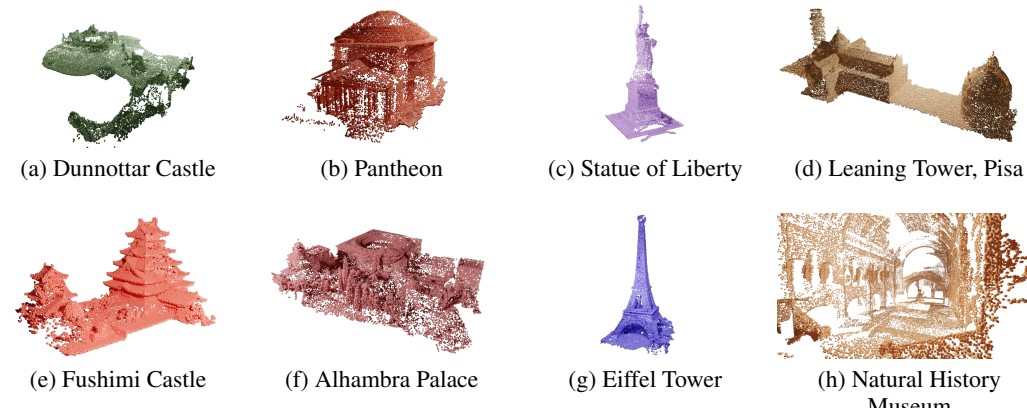

| | | | | | | | |
|---|---|---|---|---|---|---|---|
| | (a) Dunnottar Castle | (b) Pantheon | (c) Statue of Liberty | (d) Leaning Tower, Pisa | | | |
| | (e) Fushimi Castle | (f) Alhambra Palace | (g) Eiffel Tower | (h) Natural History Museum | | | |

Figure 6: **Reconstruction of large 3D scenes with our NBV approach SCONE**. We show the partial point cloud gathered by the depth sensor for each scene.

| | 3D scene | | | | | | |
|---|---|---|---|---|---|---|---|
| Method | Colosseum | Statue of Liberty | Manhattan Bridge | Fushimi Castle | Dunnottar Castle | Neuschwanstein Castle | Mean |
| Random Walk | 0.308 | 0.469 | 0.405 | 0.584 | 0.355 | 0.403 | 0.421 |
| SCONE-Entropy | 0.512 | 0.681 | 0.361 | 0.802 | 0.456 | 0.538 | 0.558 |
| SCONE | **0.571** | **0.693** | **0.685** | **0.841** | **0.739** | **0.653** | **0.697** |

(a)                                                              (b)

Figure 7: **(a) AUCs of surface coverage for several NBV selection methods during active view planning experiments on several 3D scenes from our dataset.** Results are averaged on several trajectories. **(b) Surface coverage curves for the different NBV selection methods, averaged on all 13 3D scenes.** Despite being trained only on centered ShapeNet 3D models, the second module of SCONE is able to generalize to complex scenes when predicting the visibility scores.

with local neighborhood features, the size of the global point cloud can become very large. We iterate the process either 100 times to build a full path around the object in around 5 minutes on a single Nvidia GTX1080 GPU, and recovered a partial point cloud with up to 100,000 points.

Contrary to single, small-scale object reconstruction where the object is always entirely included in the field of view, we simulated a limited range in terms of field of view and depth for the depth sensor, so that the extent of the entire scene cannot be covered in a single view. Thus, taking pictures at long range and going around the scene randomly is not an efficient strategy; the sensor must find a path around the surface to optimize the full coverage. We use a ray-casting renderer to approximate a real Lidar. More details about the experiments can be found in the appendix.

We compared the performance of SCONE with simple baselines: First, a random walk, which chooses neighboring poses randomly with uniform probabilities. Then, we evaluated an alternate version of our model, which we call SCONE-Entropy, which leverages the first module of our full model to compute occupancy probability in the scene, then selects the next best view as the position that maximizes the Shannon entropy in its field of view. The comparison is interesting, since SCONE-Entropy adapts classic NBV approaches based on information theory to a deep learning framework. Figures 5 and 6 provide qualitative results. Figure 7 shows the convergence speed of covered surface by SCONE and our two baselines, averaged on all scenes of the dataset.

Despite being trained only on ShapeNet 3D models, our method is able to compute meaningful paths around the structures and consistently reach satisfying global coverage. Since we focused on building a metric for NBV computation, this experiment is simple and does not implement any further prior from common path planning strategies: For instance, we do not compute distant NBV, nor optimal paths to move from a position to a distant NBV with respect to our volumetric mapping. There is no doubt the model could benefit from further strategies inspired by the path planning literature.

| Architecture | Mean Squared Error | IoU | | Architecture | Kullback-Leibler Divergence |
|---|---|---|---|---|---|
| Base Architecture | 0.0397 | 0.843 | | Base Architecture | **0.00039** |
| No Neighborhood Feature | 0.0816 | 0.702 | | No prediction $\hat{\sigma}$ | 0.00051 |
| With Camera History | **0.0386** | **0.844** | | No Camera History | 0.00045 |

(a) Occupancy probability  (b) Coverage gain

Figure 8: **(a) Comparison of Mean Squared Error and IoU for variations of our occupancy probability prediction model. (b) Comparison of Kullback-Leibler divergence for variations of our coverage gain computation model.** Thanks to the volumetric prediction $\hat{\sigma}$, the full model predicts a better distribution of coverage gains on the whole space as it is indicated by KL divergence, which is convenient for full path planning in a 3D scene.

## 3.3 Ablation Study

We now provide an ablation study for both modules of our full model: The prediction of occupancy probability, and the computation of coverage gain from predictions about visibility gains. To evaluate both modules separately, we compared their performance on their training critera: MSE for occupancy probability, and softmax followed by KL-Div on a dense set of cameras for coverage gain. The values are reported in Figure 8. We provide extensive details and analysis in the appendix.

**Occupancy probability.** Apart from the base architecture presented in Figure 3, we trained the first module under two additional settings. First, we removed the multi-scale neighborhood features computed by downsampling the partial point cloud $P_H$, and trained the module to predict an occupancy probability value $\hat{\sigma}$ only from global feature $g(P_H)$ and encoding $f(x)$. As anticipated, the lack of neighborhood features not only prevents the model from an efficient scaling to large scenes, but also causes a huge loss in performance. We also trained a slightly more complex version of the module by feeding it the camera history harmonics $h_H(x)$ as an additional feature. It appears this helps the model to increase its confidence and make better predictions, but the gains are quite marginal.

**Visibility gain.** We trained two variations of our second module. First, we completely removed the geometric prediction: We use directly the surface points as proxy points, mapped with an occupancy probability equal to 1. As a consequence, the model suffers from a significant loss in performance to compute coverage gain from its predicted visibility gain functions. Thus, we can confirm that the volumetric framework improves the performance of SCONE. This result is remarkable, since surface representations usually make better NBV predictions for dense reconstruction of detailed objects in literature. On the contrary, we show that our formalism is able to achieve higher performance by leveraging a volumetric representation with a high resolution deep implicit function $\hat{\sigma}$. We trained a second variation without the spherical mappings $h_H$ of camera history. We verified that such additional features increase performance, especially at the end of a 3D reconstruction.

## 4 Limitations and Conclusion

Beyond the prediction of the Next Best View, our method SCONE is able to evaluate the value of any camera pose given a current estimate of the scene volume. We demonstrated this ability with a simple path planning algorithm that looks for the next pose around the current pose iteratively. However:

- Like current methods, SCONE relies on a depth sensor and assumes that the captured depth maps are perfect. In practice, such a sensor is not necessarily available and can also be noisy. It would be very interesting to extend the approach to color cameras.
- Also like current methods, it is limited to the prediction of the camera pose for the next step. This is greedy, non-optimal as multiple poses are required anyway for a complete 3D reconstruction. Thanks to its scalability, we believe that our method could be extended to the prediction of the "Next Best Path", where future camera poses would be predicted jointly for their complementarity.

## Acknowledgements

This work was granted access to the HPC resources of IDRIS under the allocation 2022-AD011013387 made by GENCI. We thank Renaud Marlet, Elliot Vincent, Romain Loiseau, and Tom Monnier for inspiring discussions and valuable feedback.

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
