# OpenReview forum: "SCONE: Surface Coverage Optimization in Unknown Environments by Volumetric Integration"
_NeurIPS.cc/2022/Conference — NeurIPS 2022 Accept_

### Official Review · Reviewer_tPcs · 2022-07-08

**Rating:** 6
**Confidence:** 3
**Soundness:** 3 good
**Presentation:** 2 fair
**Contribution:** 3 good

**Summary:**

The paper proposes a method for next-best-view prediction during three-dimensional surface reconstruction using depth sensors. A characteristic feature of the method is its scalability: most of the state-of-the-art methods are able to work only with artificial models from the ShapeNet dataset of very limited dimensions, while the proposed method is aiming to tackle large-scale structure-from-motion reconstructions such as the one of Colosseum.  The paper introduces a formal theoretical framework explaining that it is possible to estimate surface visibility metrics by sampling points in the volume around the surface, not necessarily strictly on the surface. The proposed method build on this framework. It is evaluated on ShapeNet and on a proposed dataset of large-scale models.

**Questions:**

L67: Could you please explain the meaning of “volumetric methods are less efficient ..because they dilute information in the 3D space”

L75 this maps is not -> this map could be

Rephrase introduction and abstract to emphasize the input precisely (e.g., ‘a method taking a point cloud of points observed so far as input’). Right now the introduction only tells about a probabilistic occupancy map, referring that the perfect map is not known at run-time, but not describing, what is exactly considered to be known.

L78 ‘predict unknown geometry’ - What do these words refer to?

L93: Maybe rename ‘knowledge factor’ to ‘knowledge indicator’ as it can only  be equal to 0 or 1.

In (4) dx -> dx_0

L122: the volume is supposed to be opaque - what property exactly is expected here, in more precise terms?

L177: How does the model sample 3D points using the predicted occupancy probability function?

In experiment 3.1, how are the partial point clouds generated?

L256-257: how can ray tracing be used to render a point cloud? What is done to make this rendering approximate specifically the LIDAR sensor, but not time-of-flight or active stereo depth sensors?

L292: ‘Model suffers ... to compute coverage gain’ - does it mean that the second model suffers to predict coverage gain from such a set of points?

**Limitations:**

The prediction of coverage gain is done in the 2D camera viewpoint space rather than in the 5D space

The input to the method is not clearly formalized

It is difficult to judge about the limitations of the current method in terms of accuracy. Would be nice to illustrate some cases when the method does not perform well, and reason about why it happens so.

Would be nice to see how accurate the occupancy prediction model is, how far can it extrapolate geometry, as I think it is the main property this model should have. The second model should decide where can it expect to see more geometry that was previously unseen. Essentially it means that the first model should do occupancy map completion. How well does the model perform in this taskm is a good question for an additional ablation study.



**Strengths And Weaknesses:**

The paper develops a new theoretical framework, that explains an approach chosen by the authors to sample points for surface visibility prediction. In particular, the paper shows that it is possible to sample  points in the volume around the surface rather than on the surface directly.

The paper proposes to use two models: one for occupancy prediction, and another one for point visibility estimation for a particular camera viewpoint. The first model for occupancy prediction is elegant and scalable, reminding of fully convolutional models for images, but working with point clouds. It shows an approach to build spatially localized models for occupancy prediction effectively working with local sub-sets of input points.

The approach with decomposing the problem into two models seems to be the key to achieving scalability of prediction. It is a distinct step forward in this field, because previous models assumed small object-like structures and simply predicted probabilities of choosing the next view from a pre-defined set of viewpoints evenly distributed on a sphere. The new method is significantly more powerful both in tackling arbitrary geometry and large scale models.


One of interesting questions possibly weakening the paper is that it is rather uncommon to use depth sensors for reconstructing large-scale objects. Usually people focus on Structure-from-motion or photogrammetry tools in these cases, and these tools rely on stereo algorithms rather than on the depth sensors. However, one may argue that a stereo method applied to a pair of images can be understood as a depth sensor.

In general, the theoretical framework developed in the paper is concerned with point sampling. However, when the method itself is explained, a particular approach to point sampling used in the method is not described well. This way, the theory leaves an impression of being rather disconnected from practice in this particular paper.

I see a conceptual difficulty embedded in a formula (8). However, the formula (8) defines dependency of point visibility prediction on point-camera direction. First of all, camera orientation does not enter the formula at all. It is unclear how to find the best orientation of the camera. Next, an infinite number of camera viewpoints lying on a ray have the same camera-point direction. It is unclear which viewpoint should one choose. Importantly,  occupancy status of the point with respect to the cameras lying on this ray may change. One may move along the ray further away, and if the ray intersects some surface, then the point will become occluded starting from that intersection. This seems not to be addressed by the current model. The method, as I see it, is based on a significant simplification: it downscales the space of camera viewpoints from the 5-dimensional space down to two dimensions. It would be very interesting to clarify this point.

It remains unclear, what kind of input does the method expect. Sometimes, we see a reference to a ‘depth sensor’. In the ShapeNet experiment, it remains unclear, what is given as input. I would really like to see some clarification in the paper, what particular type of depth sensor should to be used with this method.

---

> ### Author Response · Authors · 2022-08-02
> **Before answering the reviewer's questions, we would like to emphasize that the major weaknesses identified by the reviewer are due to misunderstanding. We will take the reviewer's valuable comments into account and modify our paper to improve readability.**
>
> We thank the reviewer for their valuable comments and suggestions, that will help us clarify potential misunderstanding and increase the readability of the paper.
>
> Before answering the questions, we would like to emphasize that the major weaknesses identified by the reviewer are due to misunderstanding, and that our model does not suffer from such weaknesses as explained below.
>
> **Weakness 1: I see a conceptual difficulty embedded in  formula (8). However, the formula (8) defines dependency of point visibility prediction on point-camera direction. First of all, camera orientation does not enter the formula at all. It is unclear how to find the best orientation of the camera.**
>
> Actually, formula (8) does take the camera's viewing direction into account, thanks to the term $\hat{\chi}_c$.
>
> $\hat{\chi}_c$ is the set of all occupied proxy points in the field of view of camera $c$. We define the camera field of view as a pyramidal frustum that depends on the camera orientation. For a given camera $c$, a 3D proxy point participates to the Monte Carlo Integral in Equation (8) if and only if it is in the field of view of camera $c$.
>
> Consequently, the model computes the best orientation of the camera as the orientation with the highest predicted coverage gain, i.e. the orientation that maximizes in its field of view the number of points with high predicted visibility gains.
>
> **Weakness 2: Next, an infinite number of camera viewpoints lying on a ray have the same camera-point direction. It is unclear which viewpoint should one choose.**
>
> Since formula (8) depends on the direction and field of view of the camera (see our answer above), this is incorrect: The camera viewpoints lying on a ray will not have the same predicted coverage gain, and are not equivalent. If the camera moves along the ray, the field of view will vary and points will enter or exit the field of view of the camera, which will modify the value of the predicted coverage gain.
>
> Moreover, the point-camera vector $c-p_i$ will change for the points $p_i$ located inside the camera field of view (we will add a figure to the supplementary material to better illustrate this point).
>
> **Weakness 3: Importantly, occupancy status of the point with respect to the cameras lying on this ray may change. One may move along the ray further away, and if the ray intersects some surface, then the point will become occluded starting from that intersection. This seems not to be addressed by the current model.**
>
> Actually, this occlusion problem is addressed by the current model, and is precisely the reason why we use a self-attention unit in the second module, illustrated in Figure 3.
>
> Let $p$ be a point visible in the camera field of view. If the camera moves further away and intersects some surface, the new surface points that occlude the previous point $p$ will enter the camera field of view; thus, they will participate in the Monte-Carlo integral presented in Equation (8).
>
> In particular, the second module takes as input a camera pose $c$ as well as the sequence of proxy points located in the camera field of view. Thanks to its self-attention unit, the model will understand that some points are located in front of the previous point $p$, and it will be aware of its occlusion. Consequently, the predicted visibility gain of the previous point $p$ will be lowered. In other words, our model does not process the points independently, but uses an attention mechanism to encode the occlusion effects between the points located in the current camera field of view.
>
> **Weakness 4: The method, as I see it, is based on a significant simplification: it downscales the space of camera viewpoints from the 5-dimensional space down to two dimensions. It would be very interesting to clarify this point.**
>
> The method does not downscale the space of camera viewpoints. As explained in our answers above, both the orientation and the position of a camera are taken into account when computing its coverage gain. The formula (8) is not based on any simplification: it is just a Monte-Carlo approximation of the volumetric integral presented in equation (6) of the paper, where both the occupancy map and visibility gain are predicted using neural networks. Therefore, it is very general and designed to handle free camera motion on a 5D or even a 6D grid if needed, depending on the representation used for camera rotations.
>
> In the supplementary material, we provide images as well as videos of trajectories in large 3D scenes that illustrate how the model is able to choose the best positions and orientations on a 5D grid to incrementally build a meaningful trajectory that consistently covers most of the scene's surface. We also provide images of the point clouds gathered by the sensor along the trajectories.

---

> ### Author Response · Authors · 2022-08-02
> **We answer the first questions asked by the reviewer.**
>
> In this second comment, we would like to answer the first questions asked by the reviewer.
>
> **Q1: It remains unclear, what kind of input does the method expect.**
>
> At time $t$, our method takes as input:
> 1. A partial point cloud gathered by a depth sensor before time $t$ in the scene, as well as the previous sensor poses. We will explain the nature of the sensor in the following questions.
> 2. Any camera pose $c$ that could be a candidate to the Next Best View.
>
> The partial point cloud is processed by the first module of our model to predict an occupancy probability map representing the geometry of the unknown scene.
>
> Given the predicted occupancy map, the second module of our model predicts the surface coverage gain achieved by camera $c$.
>
> We will follow the reviewer's suggestion and modify the paper to clarify the input expected by the model.
>
> **Q2: L67: Could you please explain the meaning of “volumetric methods are less efficient .. because they dilute information in the 3D space”**
>
> Since most volumetric methods rely on memory-heavy representations (like a voxelization or a 3D-grid), they are generally less efficient for encoding very fine details and optimizing dense reconstructions. As an example, to represent a 3D mesh, an occupancy grid or a volumetric voxelization will necessarily downgrade the resolution compared to a point cloud directly sampled on the surface.
>
> To address this issue while still working with a volumetric representation of the scene, we use a deep implicit function to encode the 3D mapping of occupancy efficiently. Such a function has a virtually infinite resolution, and prevents us from saving a large 3D grid in memory.
>
> **Q3: L78 ‘predict unknown geometry’ - What do these words refer to?**
>
> These words refer to the computation of the occupancy probability map, which encodes and makes predictions about the 3D geometry of the scene based on the partial point cloud gathered by the depth sensor.
>
> **Q4: L122: the volume is supposed to be opaque - what property exactly is expected here, in more precise terms?**
>
> To maximize surface coverage with depth sensors, we consider the surface to be opaque, i.e., surfaces have no transparency: A point is considered occluded as soon as another part of the surface is located between the camera and the point, and its visibility value is considered to be 0.
>
> **Q5: L177: How does the model sample 3D points using the predicted occupancy probability function?**
>
> For a given camera $c$, we first compute the occupancy probabilities of proxy points located in the camera field of view. Then, we sample a subset of these points with a Monte-Carlo sampling: We sample $N$ points with probabilities that are proportional to their occupancy values. The sequence of coordinates of the sampled points are concatenated to their occupancy values and fed to the second module represented in Figure 3.
>
> The number $N$ can be set arbitrarily high depending on the GPU memory used to run the model. In our experiments, we used $N=2048$.
>
> **Q6: In experiment 3.1, how are the partial point clouds generated? L256-257: how can ray tracing be used to render a point cloud? What is done to make this rendering approximate specifically the LIDAR sensor, but not time-of-flight or active stereo depth sensors?**
>
> The partial point clouds are generated using the same process than experiment 3.2: we use a ray-casting renderer that outputs a depth map. The renderer casts rays inside the camera field of view from the camera position; When a ray intersects the surface, the depth value is saved and projected onto the corresponding pixels in the depth map. The depth map is then backprojected to 3D to create a partial point cloud.
>
> This renderer is designed to mimic LiDAR-class sensors such as time-of-flight cameras, that use laser rays to estimate depth and output depth maps. The density of surface points gathered by these sensors vary with the angle between the normal of the surface and the direction of observation, just like stereo depth sensors. This is the most important property that we wanted to reproduce with our virtual renderer.
>
> Finally, as the reviewer mentioned, RGB cameras are widely used for 3D reconstruction (photogrammetry, structure-from-motion) from images that have already been captured by a sensor. However, when it comes to path planning strategies and NBV computation, point clouds and depth maps are mostly used as the input.

---

> ### Author Response · Authors · 2022-08-02
> **We answer the last questions asked by the reviewer.**
>
> In this third comment, we would like to answer the last questions asked by the reviewer.
>
> **Q7: L292: ‘Model suffers ... to compute coverage gain’ - does it mean that the second model suffers to predict coverage gain from such a set of points?**
>
> Exactly. When using only the points gathered on the surface by the sensor to compute coverage gain, the performance of the second module decreases. This result shows that making a geometric prediction with the first module increases performance.
>
> **Q8:  Would be nice to see how accurate the occupancy prediction model is, how far can it extrapolate geometry, as I think it is the main property this model should have. The second model should decide where can it expect to see more geometry that was previously unseen. Essentially it means that the first model should do occupancy map completion. How well does the model perform in this task is a good question for an additional ablation study.**
>
> As the reviewer suggests, the first model is designed to make occupancy map completion. However, we want our model to be as general as possible; we do not want it to overfit on specific categories of objects nor make too strong assumptions on the geometry.
>
> This is why our occupancy prediction model outputs a gradient of probabilities from 0 to 1 that helps to identify the areas that should have high or low density, as well as the areas where incertitude relies. When a single observation (i.e., a single depth map) of the scene is available, the predicted occupancy map generally identifies large areas of incertitude (i.e. with values around 0.5). However, the more depth maps are gathered, the better the prediction becomes, even for unknown environments or categories of objects never seen by the model.
>
> The second model is indeed trained to decide where it can expect to see more geometry that was previously unseen.

---

### Official Review · Reviewer_CVjA · 2022-07-09

**Rating:** 7
**Confidence:** 3
**Soundness:** 3 good
**Presentation:** 2 fair
**Contribution:** 3 good

**Summary:**

This paper aims to solve the problem of next best view from partial point cloud towards a complete reconstruction. The method uses the principle of SDF  and define a way of computing the maximum coverage gain if a camera is at position "c" given the history of camera position, the visibility of the surface on those historical position. This in turn  expected to maximise the total coverage. The authors derive the relations of the incremental coverage gain mathematically and use a neural network to realise those relation to produce the coverage gain given a camera position "c". They show results on various dataset including large scale reconstruction.

**Questions:**

1. Isn't the surface point could be obtained using ray intersection from camera in line 100? Or am I missing something?
2. Its not clear if the point cloud has a noise how it impact the visibility gain and occupancy function? Though authors mention this in the limitation, but the discussion on sensitivity of noise could have been good.
3. If the ablation is given with the reconstruction quality then it will be more clear to the reader about the significance of the numbers.
4. How the authors create the ground truth of the surface coverage gain?
5. If the explanation from Eq. 6 to the neural design is more lucidly written, then the readability will increase.
6. Is the method tried in indoor? How this method is positioned w.r.t https://arxiv.org/abs/1805.07794
7. Is it possible to use the Pointnet features for encoding the 3D point and its neighbourhood in Fig 2?
8. How the occlusion is handled?

**Ethics Review Area:**

["I don’t know"]

**Limitations:**

The limitation of this method is narrated by authors and some of the insight we can get from the above questions. For reproducibility, the crucial information regarding the training needs to be there in the main paper.

**Strengths And Weaknesses:**

Strengths:
The paper has a strong mathematical foundation on defining the coverage gain. The use of spherical harmonics for visibility gain is nice, though SH is being used in recent volume rendering work which authors also referred. Supplementary method provides all the derivation required for the proof of the relations used in the main paper. The paper shows interesting results.

Weaknesses
The paper is well grounded with the theoretical formulations. But the readability of the  paper is not very good. It is relatively difficult to relate the equations with the neural architecture for a reader. The concepts are philosophically mapped but the derivation is not clearly mapped with the neural method or in the inductive bias design. The paragraph from line 132 to136 attempted to give some clarity but this needs to be expanded for a general reader. Its not clear if there were neural aggregator instead of Monte Carlo, then what could happened? In table 1 the numbers are very close. What is the sensitivity of these numbers w.r.t the reconstruction?

---

> ### Author Response · Authors · 2022-08-02
> **We answer the Questions 1, 2, 3, 4, 6 and 7 asked by the reviewer.**
>
> We thank the reviewer for their valuable comments and suggestions, and would like to answer their questions.
>
> **Q1: Isn't the surface point could be obtained using ray intersection from camera in line 100? Or am I missing something?**
>
> This would indeed work, but only if the occupancy map was known perfectly. In our case, we estimate the occupancy map iteratively, and we only have access to the occupancy probability distribution.
>
> This is why we compute a volumetric integral of visibility gains instead of extracting surface points from our predicted occupancy maps. In 3D, a surface acts as a very concentrated set (with zero-measure), and requires high confidence to give meaningful results. On the contrary, our model outputs a gradient of probabilities that helps to identify the areas that should have high or low density, as well as the areas where incertitude relies. Extracting surface points from such a probabilistic occupancy map gives results that can differ a lot from the true surface. Instead, we found that a volumetric integration of visibility gain on the whole occupancy distribution was more efficient to make accurate predictions about geometry and NBVs. We will update the paper to clarify this point.
>
> **Q2: It's not clear if the point cloud has a noise how it impact the visibility gain and occupancy function? Though authors mention this in the limitation, but the discussion on sensitivity of noise could have been good.**
>
> Such a discussion on sensitivity of noise has been conducted by Zeng *et al.* [33] on their model PC-NBV: They added a small Gaussian noise to the coordinates of the gathered partial point cloud for ShapeNet models, and found out that the model was quite resistant to noise. We actually conducted the same experiment, that led to similar results.
>
> In the paper, we were referring to the noise and imperfections that exist in depth maps captured by real depth sensors. In other words, we wonder how our model would handle the domain gap between synthetic and real data. Unfortunately, we were not able to make experiments with a real UAV.
>
> **Q3: If the ablation is given with the reconstruction quality then it will be more clear to the reader about the significance of the numbers.**
>
> In the supplementary material, we provide in figure 2 the evolution of surface coverage throughout reconstruction of small scale objects by our model SCONE as well as several other methods. We provide similar curves for 13 large 3D scenes in figure 5 of the supplementary material, as well as examples of reconstructions of the same 13 scenes in figure 4.
>
> **Q4: How the authors create the ground truth of the surface coverage gain?**
>
> The ground truth of surface coverage gain is computed from ground truth surface points following Equation (12) in the supplementary material. To compute ground truth surface points, we sample points on the ground truth mesh triangles and make sure that the sampling follows an uniform distribution on the whole surface.
>
> **Q6.1: Is the method tried in indoor?**
>
> We provide an image and a video in the supplementary material of the trajectory retrieved by our method for an indoor scene (the London Natural History Museum).
>
> **Q6.2: How this method is positioned w.r.t https://arxiv.org/abs/1805.07794**
>
> The reference mentioned by the reviewer is very interesting, but specializes to indoor scenes where the objects belong to categories already known by the system. By contrast, our approach is much more general, as it does not need a data bank of object models and it is not object-centered.
>
> **Q7: Is it possible to use the Pointnet features for encoding the 3D point and its neighbourhood in Fig 2?**
>
> The most important part of the occupancy prediction module is the computation of local neighbourhood features, and the type of encoders used to process the points have little importance. Therefore, the reviewer is entirely right, Pointnet features could be used.  Actually, we tried Pointnet features trained from scratch to encode the 3D points during our development, but the self-attention units turned out to be slightly more efficient.
>
> **Additional Question: In table 1 the numbers are very close. What is the sensitivity of these numbers w.r.t the reconstruction?**
>
> For this experiment, we constrain the camera to stay on a sphere centered on small scale objects. Thanks to this experiment, we can compare our model with previous methods trained for the very specific case of dense object reconstruction with camera motion constrained on a sphere.
>
> The point of this experiment was to prove that, even if our model is designed to handle entire scene reconstruction with free camera motion, it is still able to beat other methods trained for this specific case.
>
> Therefore, the fact the numbers in table 1 are close is not a problem, since the main strength of our model is its scalability to free camera motion and arbitrarily large 3D scenes.

---

> ### Author Response · Authors · 2022-08-02
> **We answer the remaining Questions 8 and especially 5, about the link between Equation 6 and the neural design of our model.**
>
> In this second comment, we would like to answer the remaining questions 8 and 5 asked by the reviewer.
>
> **Q8: How the occlusion is handled?**
>
> Occlusions are handled by a self-attention unit, as shown in red in Figure 3 of the paper. For a given camera $c$, we first compute the occupancy probabilities of proxy points located in the camera field of view using the first module of SCONE.
>
> Then, we sample a subset of these points with a Monte-Carlo sampling: Each point has a sampling probability proportional to its occupancy value.
>
> The coordinates of the sampled points are concatenated to their occupancy values and fed to a small MLP, shared between all points.
>
> The sequence of resulting features is then fed to the self-attention unit. This unit encodes the interaction between all points. Indeed, for each point $p$, the output of the self-attention unit encodes implicit information about the other points surrounding $p$, their occupancy, and which direction could be occluded.
>
> The resulting feature is concatenated to an additional feature that encodes the relative location of previous camera poses, and finally fed to another shared MLP that outputs coordinates in spherical harmonics representing visibility gains.
>
> **Q5: If the explanation from Eq. 6 to the neural design is more lucidly written, then the readability will increase. [...] Its not clear if there were neural aggregator instead of Monte Carlo, then what could happened?**
>
> To approximate the volumetric integral in Equation (6) for any camera pose $c$, we need to compute $\chi_c$ as well as function $g_c^H$. In this regard, we need to compute both the occupancy map and the visibility gains of points for any camera pose. Since the environment is not perfectly known, we predict each one of these functions with a dedicated neural module.
>
> The first module of our model directly takes as input the partial point cloud gathered by the depth sensor to predict the occupancy probability distribution $\hat{\sigma}$. To predict the occupancy map of an arbitrarily large scene, we need the model to be scalable. This is why we designed our first module around neighborhood features, rather than a single global encoding of the whole scene's geometry.
>
> Then, the second module takes as input a camera pose, a feature representing camera history (i.e. the previous camera positions) as well as a sequence of proxy points located in the camera field of view to predict the visibility gains and the resulting coverage gain of the camera. Proxy points are sampled using the predicted occupancy probability distribution $\hat{\sigma}$. We use a self-attention unit on the sequence of points located in the camera field of view to encode the occlusion effect between the points.
> We use spherical harmonics to encode camera history. The output visibility gains are also sets of coordinates in spherical harmonics, so that we can predict the gain for several cameras in the same time if they share the same points in their field of view. Moreover, it makes the output very homogeneous to the input camera history, and helps the model to achieve faster convergence and better performance.
>
> To aggregate the per-point visibility gains to approximate the integral in Equation (6), we choose to use a Monte-Carlo integral rather than a neural aggregator. In this regard, Equation (8), which represents the output computed by our model, is actually just a Monte Carlo approximation of the volumetric integral of Equation (6). This approach is simple, fast, makes training more stable, has good performance, better interpretability, and can handle sequences of arbitrary size. In particular, it implicitly encourages our model to compute meaningful visibility gains for each point since there is no asymmetry between the points. As we explained, we sample the 3D points in the camera field of view with probabilities proportional to their occupancy values, just as we expect with a MC integration.
>
> Please note that the unknown variable $\mu$ appearing in $g_c^H$ in Equation (6) is not explicitly fed to the networks, but is implicitly handled by the model; the only inputs to the full model are the partial point cloud gathered by the depth sensor and the camera poses.
>
> To make the results reproducible, we will release code and also add details in the supplementary material about the neural design, in particular about the number and size of the layers in the different MLP and SA units.

---

> > ### Comment · Reviewer_CVjA · 2022-08-07
> > **After Rebuttle**
> >
> > Thanks to the authors for the rebuttle. Many of the doubts are clear. Please include the monte carlo and neural aggregator discussion briefly in the paper.

---

### Official Review · Reviewer_DgoJ · 2022-07-09

**Rating:** 8
**Confidence:** 4
**Soundness:** 4 excellent
**Presentation:** 4 excellent
**Contribution:** 4 excellent

**Summary:**

This paper describes a method for next-best view prediction for reconstructing large-scale 3D scenes with a depth sensor.  They derive a formula to estimate the surface coverage gain for any potential camera pose given a camera pose history and a probabilistic occupancy map.  They use one neural network to predict probabilistic occupancy map based on a point cloud input, and a second network to predict the visibility gain, which is used in the calculation of surface coverage gain.  The first network is trained to match the ground truth occupancy map, and the second is trained to match the ground-truth surface coverage gain.  Extensive experiments on synthetic datasets demonstrates an improvement in performance over SOTA.

**Questions:**

L89: the expression (1- \lambda) c_pos + \lambda x seems to be interpolating between the camera position and the point, but it doesn't take the camera's viewing direction into account?
L101: "tubular" -- is this the right word?  or should it be "spherical"?  I couldn't figure out why the neighborhood region would be tubular.  Based on the expression in L105 it seems to be spherical (all points within a distance of \mu_o from x.



**Limitations:**

Limitations are discussed but not potential negative societal impacts.

**Strengths And Weaknesses:**

They provide a thorough derivation an approximation of surface coverage gain that they prove asymptotically approaches the true value.  This formulation is novel to the best of my knowledge.

The structure of the neural network and the loss functions is interesting.  They use separate networks to predict the probabilistic occupancy  map and the visibility gain functions.  The loss function for the visibility gain network is not visibility gain itself but the surface coverage gain.  They also use attention mechanisms to model occlusion effects.

They provide a thorough set of experiments to demonstrate their method leads to an improvement on ShapeNet following a standard protocol.

They also provide some ablation studies to establish the usefulness of various aspects of their proposed approach.

Overall, they have some interesting new ideas which lead to an increase in performance for NBV selection, and their work also can lead to new research such as handling noise in the depth map and selecting optimal paths rather than single viewpoints.

---

> ### Author Response · Authors · 2022-08-01
> **We answer the reviewer's questions and propose to change some notations that can be misleading.**
>
> We thank the reviewer for their valuable comments and suggestions, and would like to answer their questions.
>
> **Q1: L89: the expression $(1- \lambda) c_{pos} + \lambda x$ seems to be interpolating between the camera position and the point, but it doesn't take the camera's viewing direction into account?**
>
> A1: Actually, the complete expression $\mathbb{1}_{\chi_c}(x) \cdot \mathbb{1}\left(\sigma\left(\{(1-\lambda) c_\text{pos} + \lambda x \text{ such that } \lambda\in[0,1)\}\right)=\{0\}\right)$ does take the camera's viewing direction into account, thanks to the term $\chi_c$: $\chi_c$ is the set of all occupied points in the field of view of camera $c$. For a given camera $c$, a 3D point participates to Integral (1) if and only if it is in the field of view of camera $c$ and if there is no other point that occludes it.
>
> We agree however the current notation can be misleading. We propose to move the indicator function directly in the definition of the visibility $v_c$ to avoid misunderstandings; we will change the definition of $v_c$ accordingly, to $v_c:x \mapsto  \mathbb{1}_{\chi_c}(x) \cdot \mathbb{1}\left(\sigma\left(\{(1-\lambda) c_\text{pos} + \lambda x \text{ such that } \lambda\in[0,1)\}\right)=\{0\}\right)$.
>
> **Q2: L101: "tubular" -- is this the right word? or should it be "spherical"? I couldn't figure out why the neighborhood region would be tubular. Based on the expression in L105 it seems to be spherical (all points within a distance of $\mu_0$ from x**
>
> A2: The reviewer is right, "spherical" fits well the expression in L105. We used the word "tubular" because it was used in particular in [10] (Gilbarg *et al*.), where Equation (4) comes from.
>
> Tubular neighborhoods are specific neighborhoods of submanifolds resembling the normal bundle. Equation (4) actually only applies to tubular neighborhoods; However, the spherical neighborhoods of $C^2$ watertight surfaces also are tubular neighborhoods. Since we use both properties of tubular and spherical neighborhoods in our proof but wanted to keep the definitions as simple as possible, we used the definition of spherical neighborhoods.
>
> To make the proof less confusing, we propose to follow the reviewer's suggestion and change the word "tubular" to "spherical". As a consequence, we will add a comment in the supplementary material to explain that the spherical neighborhoods we use also are tubular.

---

### Meta-Review · Area_Chair_UT25 · 2022-08-24

**Recommendation:** Accept
**Confidence:** Certain

**Metareview:**

The paper describes an approach to next-best-view (NBV) planning for the reconstruction of large-scale 3D scenes using depth sensors. The proposed framework models the scene using a probabilistic occupancy map and chooses the next-best-view as the free camera pose that maximizes the gain in surface coverage. Integral to the approach's ability to handle large-scale scenes is the paper's formulation of surface coverage estimation as sample-based volumetric integration. Based on this formulation, the approach employs one neural network to predict the visibilities that are used to calculate surface coverage gain, and a second network to estimate the probabilistic occupancy map from the point cloud input. The paper presents experimental evaluations on the benchmark ShapeNet dataset as well as a proposed large-scale dataset, demonstrating gains over contemporary methods.

The paper was reviewed by three reviewers who read the author response and discussed the paper with the AC. The reviewers agree that the proposal to approximate surface coverage via sample-based volumetric integration, which is integral to the approach, is both novel and principled. To that end, the reviewers appreciate that the proposed architecture is well grounded in rigorous theoretical foundations. The experimental evaluation is thorough, with ablations that clearly demonstrate the advantages of the proposed architectural components. A key concern raised by several reviewers is that the readability of the submission is poor, which makes it difficult to relate the formal derivations to the neural network architecture. This lack of clarity lead to notable misunderstandings on the part of at least two reviewers. During the discussion phase, the reviewers acknowledged that the author response largely resolves this concern, but it is critical that the paper be updated to address these issues as well.

**Award:**

No

---

### Decision · Program_Chairs · 2022-09-14

Accept